# Study on Infiltration and Soil Moisture Characteristics of a Sand-Covered Slope



**Feichao Wang [1], Guoce Xu [1],\*, Zhanbin Li [1,2], Peng Li [1,2], Tian Wang [1], Jianwen Zhang [3], Jie Wang [4] and Yuting Cheng [5]**

[1] State Key Laboratory of Eco-Hydraulics in Northwest Arid Region of China, Xi'an University of Technology, Xi'an 710048, China; wangfeichao_0120@163.com (F.W.); zhanbinli@126.com (Z.L.); lipeng74@163.com (P.L.); t.wang@xaut.edu.cn (T.W.)

[2] Key Laboratory of National Forestry Administration on Ecological Hydrology and Disaster Prevention in Arid Regions, Xi'an University of Technology, Xi'an 710048, China

[3] China Jikan Research Institute of Engineering Investigations and Design Co., Ltd., Xi'an 710043, China; zhangjianwen95@126.com

[4] Yulin Institute of Foresty, Yulin 719000, China; wangjiejie6666@163.com

[5] College of Life Science, Yan'an University, Yan'an 716000, China; chengyutingstar@163.com

\* Correspondence: xuguoce_x@163.com; Tel./Fax: +86-29-8231-2658

**Abstract:** By observing the processes of infiltration, flow generation, water flow characteristics and the spatial distribution of erosion in a designed rainfall test, and analyzing the infiltration, flow generation characteristics, water content change characteristics, soil moisture parameters change characteristics, and changes in the spatial patterns of erosion and sediment yield, this study draws the following conclusions: under different rainfall densities, the initial runoff generation time of sand-covered slope is 1~12 min longer than that of loess slope, the initial soil infiltration velocity of sediment-covered slope is about 1.23 times that of loess slope, and the time to reach stable infiltration of loess slope is shorter than that of sediment-covered slope. Under different rainfall densities, the rising time of the water content curve of sand-covered slope is earlier than that of loess slope. For the same duration of rainfall, the vertical infiltration performance of soil water of sand-covered slope is higher than that of loess slope, and when the rainfall density on the slope increases by 0.5 mm/min, the increase in runoff shear stress of the sediment-covered slope is about 1.5 times that of the loess slope, and the runoff power is about 1.13 times that of the loess slope.

**Keywords:** sand-covered slope; soil erosion; infiltration; runoff sediment; soil moisture characteristics





## 1. Introduction

The Loess Plateau is one of the regions characterized by the most serious soil erosion in China. The annual precipitation in this region is less than in surrounding areas, the ecological environment is poor, and the vegetation recovery is diminished [1–4]. Due to erosion, a greater amount of sediment particles is transported to the Yellow River via runoff and deposited in the lower reaches, resulting in the elevation of the riverbed, and posing a serious threat to the ecological security of the lower reaches [5]. Therefore, the prevention and control of soil erosion has become the focus of the response to these global environmental issues [6–8]. Due to the variation in the physical characteristics, infiltration [9], water conductivity, and water holding capacity between the surface sand layer and the yellow soil layer, a sand interface is formed, which in turn forms the typical dual structure of sand [10–13].

Water infiltration is an important component of the hydrological cycle [14–17], as it directly determines the generation time and size of slope runoff, and also affects soil moisture at different depths [18,19]. During this process, rainfall leads to changes in the soil infiltration capacity on the slope due to sand coating, which further changes the runoff

on the slope [20]. In this way, the runoff depth increases, resulting in changes in hydraulic characteristics [21]. Previous studies have investigated this infiltration process, the runoff and sediment yield process of sand-covered slopes, and their relationship, as well as the influence of sand grain size composition on the erosion process [4,10,11,22]. In a field rainfall experiment, Zhang et al. [23] found that the runoff process of sandy slopes was obviously different than that of loess slopes. Here, the runoff of sandy slopes decreased, but the sediment content in the runoff increased. Similarly, Wu et al. [24] qualitatively described the interfacial flow of sandy soil on sandy slopes through field investigation. Additionally, through an indoor simulation study, Tang and Su [22,25] found that the initial runoff time of sandy slopes increased, and, with increasing sand thickness, under different treatments, the cumulative sediment yield increased with the increase in runoff.

Slope runoff velocity is one of the most important soil moisture parameters, and has great influence on soil erosion. Slope runoff and sediment movement are closely related to hydraulic parameters [26]. The main factors affecting the hydraulic characteristics of slope are slope sand covering, soil freezing, and so on [27–29]. Previous work has observed that the flow pattern of water has a great influence on the erosion of sandy slopes. Similarly, it has been determined that under different treatment conditions, the two parameters describing the runoff process are runoff velocity and the hydraulic parameters [27,30].

Therefore, through an indoor simulated rainfall experiment, taking the loess and sand-covered slopes in the east willow ditch in northern Shaanxi as the research object, the infiltration process, runoff characteristics, and influencing factors of the sand-covered slope were analyzed, the variation process of the soil moisture parameters was studied, and the relationship between erosion sediment yield parameters and soil moisture parameters was explored. Specifically, slope soil moisture content, time units for different parameters of runoff and sediment yield, slope runoff velocity and water depth, combined with the spatial and temporal variations of soil moisture content, were obtained via the design of different rainfall densities.

## 2. Materials and Methods

### 2.1. Test Material

Simulated rainfall experiments were conducted in the State Key Laboratory of Eco-Hydraulic Engineering at Xi'an University of Technology in China. In this test, a side jet rainfall simulation device was used, with uniformity >90%. The simulated rainfall test used a wooden soil bin that was 2 m long, 0.75 m wide, and 0.60 m high. The wood was 3 cm thick, which meant that the soil could be kept warm and that a one-dimensional thaw occurred in the vertical direction of the soil. The lower end of the soil bin was connected to a collecting tank, which was used to collect runoff and sediment samples (Figures 1 and 2). Soil moisture content was measured using the CR1000 data acquisition device from Campbell Company in the United States. Measurements of soil moisture content were obtained using a CS616 soil moisture sensor at a frequency of 1 measurement/min. Five water probes were positioned along the vertical direction at depths of 3 cm, 6 cm, 9 cm, 14 cm and 22 cm from the soil surface, respectively.

The experimental soil was selected from an alfalfa field in Wangmaogou, Suide, Northern Shaanxi Province. The surface soil was 20–30 cm in depth, and the sandy soil was characteristic of the aeolian sand of the Dongliugou watershed. After the soil samples were returned to the laboratory, debris such as grass roots and stones were removed before being passed through 10 mm (soil) and 0.8 mm (sand) sieves for pretreatment. Prior to filling the soil trough, the wooden soil trough was first soaked with flowers to allow for better bonding with the soil. In addition, a layer of fine sand was laid at the bottom of the soil groove, and the bottom of the soil groove was drilled so that the water can penetrate normally after reaching the deep layer. The soil bulk density was controlled at about 1.3 g/cm$^3$, and the soil water content was about 15%. The thickness of the soil was 40 cm, and the method of layered filling and artificial compaction was used to fill the soil. The thickness of each layer was about 5–8 cm. After each layer was filled, the surface of the soil

layer was haired to closely combine the two layers of soil. After filling the soil tank, the soil surface was covered in 2 cm of sand according to the experimental design. The adhesion to the surface soil was noted during the sand-covering, so the method of burring and sand-covering followed by compaction was adopted. Finally, in order to ensure that the test conditions of each test were consistent, the slope was leveled every time, and non-invasive rainfall with rainfall density of 30 mm/h was applied before the slope rainfall, after which the test soil tank was covered with plastic cloth for a waiting period of 24 h before testing. The characteristics of soil particles were measured using a Malvin 2000 (Malvern Worcs, WR141XZ, UK). The characteristics of the soil particles are shown in Table 1.

**Table 1.** Soil particle characteristics of test soil.

| Experimental Soil | Bulk Density (g/cm³) | Initial Soil Moisture | Clay Particle (≤0.002 mm) | Powder Particle (0.002~0.02 mm) | Sand Grains (≥0.02 mm) |
|---|---|---|---|---|---|
| Loess | 1.3 | 15% | 0.02% | 65.28% | 34.70% |
| Sand | 1.3 | 15% | 0.72% | 14.38% | 84.90% |

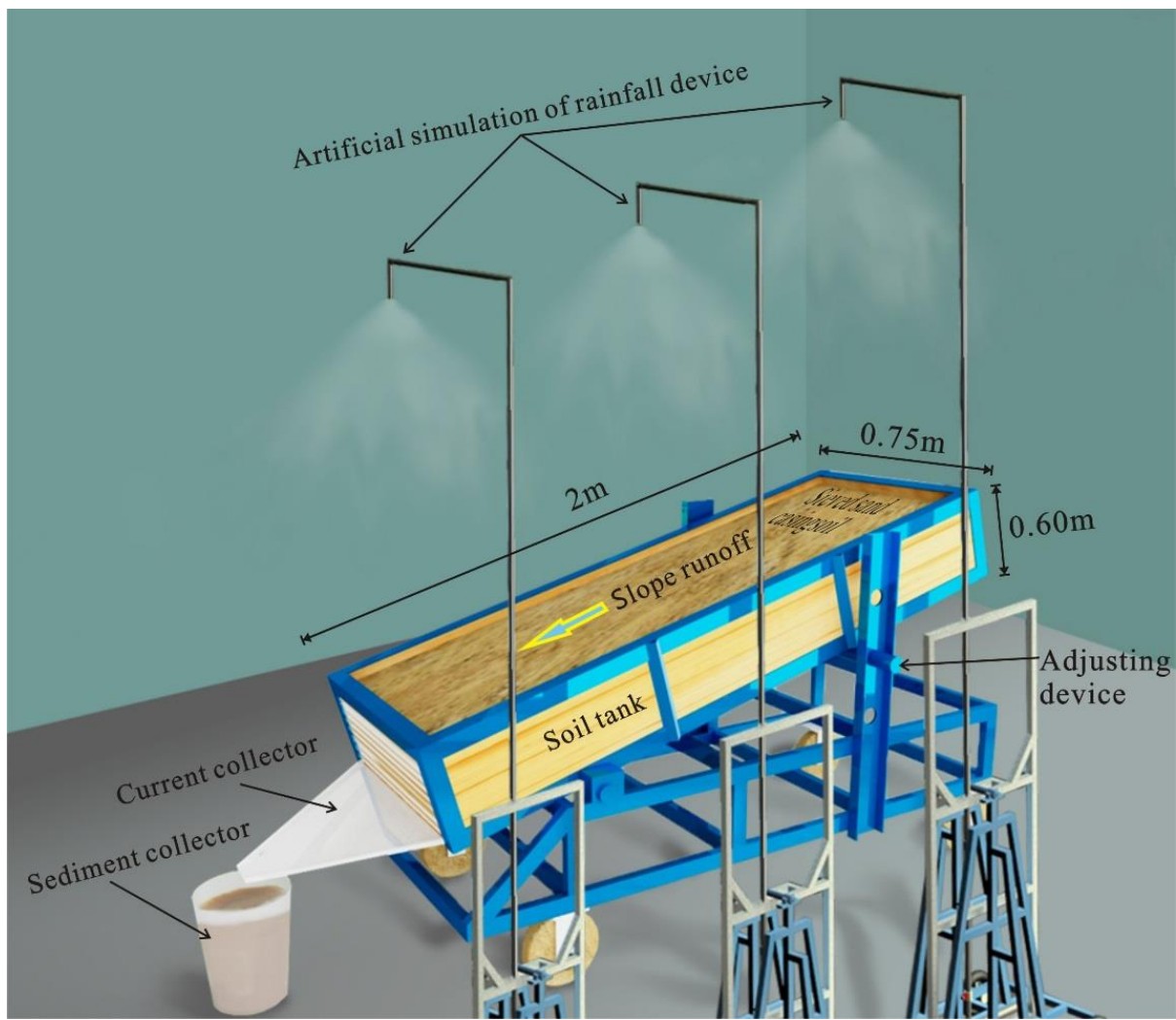

**Figure 1.** Schematic diagram of test soil trough.

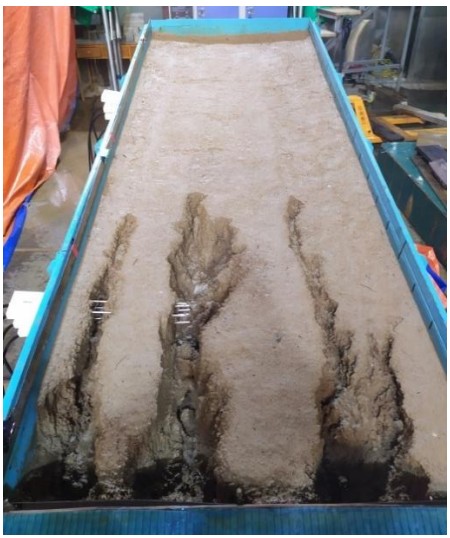

**Figure 2.** Field test photos.

*2.2. Experimental Design*

This experiment follows a simulated rainfall test design. According to previous literature and the investigation of the thickness of a sand layer deposited in the water–wind erosion crisscross region, the thickness of sand cover was determined to be 2 cm. Through the study of a simulated rainfall experiment, Zhou et al. [28] found that the standard of the loess plateau erosive rainstorm intensity was 10.50~234.84 mm/h, so the rainfall densities in this experimental design were set to 1.0, 1.5, 2.0 and 2.5 mm/min. The treatment slope surface was 2 cm sand-covered loess, and the control slope was loess slope. The slope of the test soil groove was fixed at 12°. Intermittent rainfall was applied 3 times per treatment, where each rainfall lasted for 60 min, and each rainfall interval was 24 h, which was the interval when using a rain shelter cover soil trough. The test was repeated 2 times under each condition, for a total of $4 \times 2 \times 3 \times 2 = 48$ field tests. The design scheme for the test is shown in Table 2.

**Table 2.** Design scheme of indoor rain test.

| Slope Type | Rainfall Density (mm/min) | Thickness of Sand-Covered (cm) | Rainfall Event |
|---|---|---|---|
| Sand-covered slope | 1.0 | 2 cm | 3 |
| | 1.5 | 2 | 3 |
| | 2.0 | 2 | 3 |
| | 2.5 | 2 | 3 |
| Slope land | 1.0 | 0 | 3 |
| | 1.5 | 0 | 3 |
| | 2.0 | 0 | 3 |
| | 2.5 | 0 | 3 |

*2.3. Measurement and Calculation of Indicators*

(1)  Determination of Runoff Sediment Index

Before the test, the slope surface was divided into four 0.5-m-long observation sections from bottom to top. Once the rainfall began, a 5000 mL measuring tube was placed at the runoff and sediment outlet of the soil trough. As the runoff began on the slope, the measuring cylinder was used to collect the runoff sediment samples every minute, and conical bottles were used to collect the runoff sediment samples. Each conical bottle was then left to rest for about 2 h to allow for the sediment to settle, after which the supernatant was carefully removed. The remaining runoff sediment samples were then poured into an

iron box with known weight, and placed in an oven to be dried at 105 °C prior to weighing. The sediment weight in the conical bottle was then obtained. Next, the sediment weight in the measuring cylinder was calculated by the replacement method [31]. The sum of the two measurements is the total sediment yield in this time, and the sum of the runoff in the measuring cylinder and the runoff in the conical bottle is the total runoff in this time.

(2) Calculation of runoff velocity and soil moisture parameters

- Runoff velocity

  During the experiment, a potassium permanganate staining method was used to determine the surface runoff velocity, *Vs*, of the slope runoff. The average runoff velocity of slope runoff was calculated as follows:

$$V = V_s \times \beta \tag{1}$$

where *V* is the average velocity of runoff and $\beta$ is the correction coefficient of runoff velocity, and is taken as 0.75 in this study [32].

- Runoff depth

  Since the slope flow in the experiment is a thin-layer flow, it is difficult to measure the runoff depth *h*. For this reason, previous researchers have adopted the assumption that the slope flow is uniformly distributed. The method for calculating the runoff depth in this study is as follows:

$$h = Q/(V \times B \times T) \tag{2}$$

where *h* is runoff depth, m; *Q* is the runoff during a period of *T*, m$^3$; *V* is slope average velocity, m/s; *B* is water width, m; and *T* is the duration, s. Runoff depth in rills is measured directly using the ruler method.

- Flow shear stress

  The shear force of runoff can peel soil particles from their original position by damaging the original structure of the soil and removing them from the slope with the flow. In practical research, the movement form of slope flow is simplified as a one-dimensional uniform flow. The method for calculating runoff shear stress is as follows:

$$\tau = \rho g R S \tag{3}$$

where $\tau$ is the runoff shear stress on the slope, Pa; $\rho$ is the density of rain water, 1000 kg/m$^3$; *g* is the gravity acceleration, which is 9.8 m/s$^2$; *R* is the hydraulic radius, where the hydraulic radius of the thin-layer flow is equivalent to its runoff depth, and the runoff depth of the rill is measured by a ruler, m; and *S* is the hydraulic gradient, which is simplified as the sinusoidal value of the soil groove gradient, namely $S = \sin\theta$, and $\theta$ is the soil groove gradient of 12°.

- Runoff power

  Runoff power refers to the change rate of water potential energy with time per unit of area. This study uses the following method to calculate runoff power:

$$\omega = \tau V = \rho g R S V \tag{4}$$

where $\omega$ is the runoff power of the slope, N/(m·s), and the other letters have the same meaning as denoted previously.

*2.4. Data Processing and Analysis*

In each rainfall experiment, the following data were collected: runoff generation time under two kinds of slopes, runoff velocity of slopes and rills at different rainfall times, soil moisture content, ditch width, ditch depth, and runoff per time unit.

Pearson correlation analysis was used to analyze the correlation of the collected data. Statistical and regression analyses were performed in SPSS 22.0 (Stanford University, Stanford, CA, USA) using Origin 2017 (Originlab, Northampton, MA, USA), 3ds Max (Autodesk Corporation, San Rafael, CA, USA), and Excel 2010 (Microsoft Corporation, Redmond, WA, USA).

## 3. Result

### 3.1. Infiltration and Runoff Process on Sandy Slope

As can be seen in Figure 3, the initial runoff times of the sand-covered slope was 12 min, 7 min, 3 min and 2 min under rainfall densities of 1.0, 1.5, 2.0 and 2.5 mm/min, respectively. Prior to the generation of runoff on the slope, rainfall was used for infiltration. After runoff generation on the slope, the infiltration velocity gradually decreases with the extension of rainfall time, and finally stabilizes. The stable infiltration velocity of intermittent rainfall on the slope under the four rainfall densities varied. Specifically, under the rainfall densities of 1.0, 1.5, 2.0 and 2.5 mm/min, the stable infiltration velocities of the first rainfall were 0.34, 0.2, 0.58 and 0.41 mm/min, respectively. Similarly, the stable infiltration velocities of the second rainfall were 0.32, 0.30, 0.38 and 0.31 mm/min, respectively. Finally, the stable infiltration velocities of the third rainfall were 0.21, 0.28, 0.22 and 0.20 mm/min, respectively.

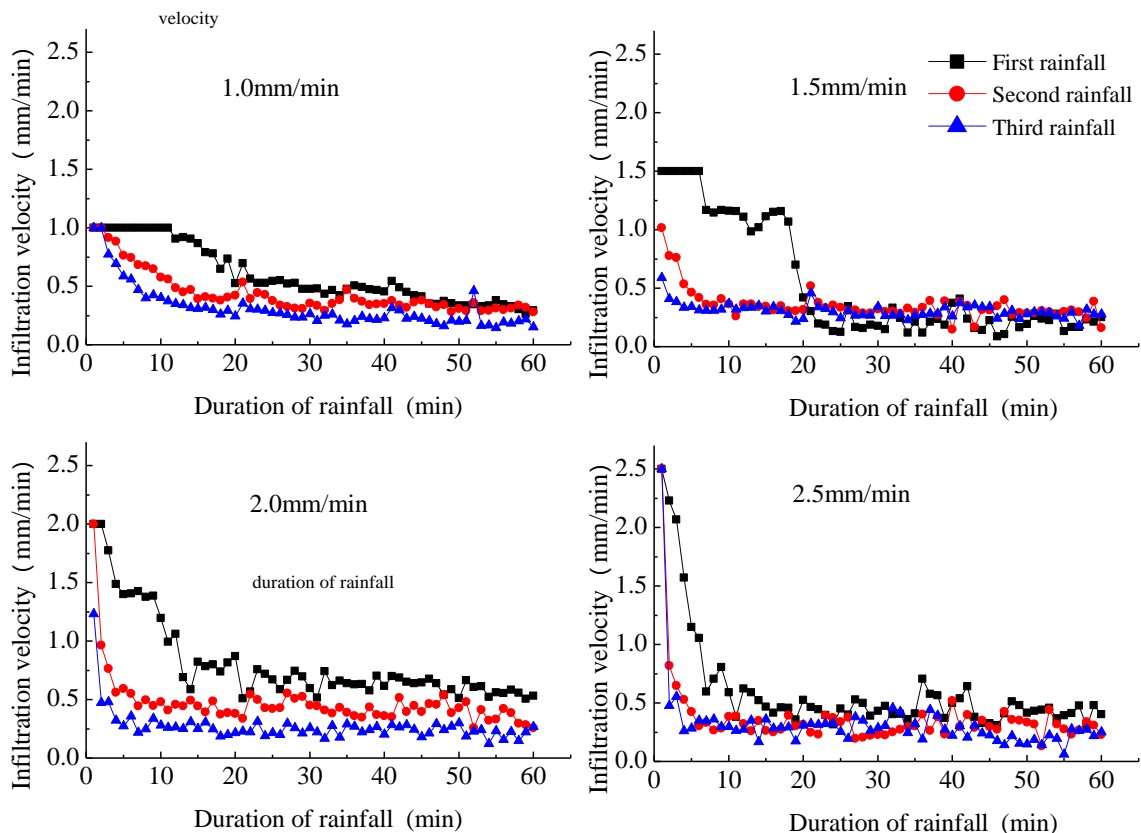

**Figure 3.** Infiltration velocity of the sand-covered slope.

It can be seen from Figure 4 that the initial runoff times of the loess slope were 3 min, 1 min, 1 min and 1 min under rainfall densities of 1.0, 1.5, 2.0 and 2.5 mm/min, respectively. After runoff generation of the loess slope, the soil infiltration velocity gradually decreased with increasing rainfall duration, until finally reaching a stable infiltration velocity. The stable infiltration velocities of intermittent rainfall on the slope under the four rainfall densities were different, with the stable infiltration velocities of the first rainfall under rainfall densities of 1.0, 1.5, 2.0 and 2.5 mm/min being 0.27, 0.32, 0.35 and 0.57 mm/min,

respectively. The stable infiltration velocities of the second rainfall were 0.26, 0.34, 0.25 and 0.33 mm/min, respectively. The stable infiltration velocities of the third rainfall were 0.23, 0.32, 0.23 and 0.28 mm/min, respectively. In the first rainfall, greater rainfall densities were associated with greater stable infiltration velocity of the soil. Similarly, the soil infiltration velocity decreased with increasing rainfall.

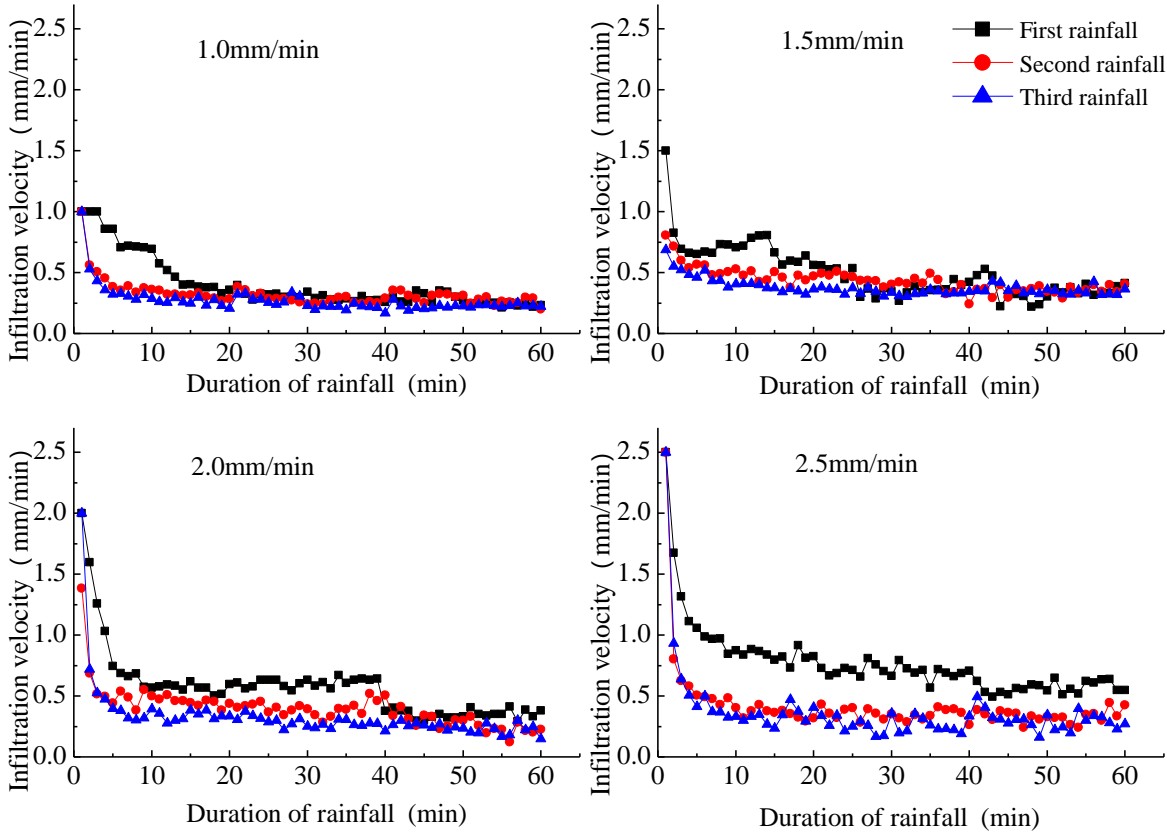

**Figure 4.** Infiltration velocity of the loess slope.

*3.2. Spatial and Temporal Distribution Characteristics of Soil Moisture on Sand-Covered Slope under Different Rainfall Densities*

3.2.1. Spatial and Temporal Distribution Characteristics of Soil Moisture on the Slope Surface under a Rainfall Density of 1.0 mm/min

Figures 5 and 6 show the dynamic change process of the soil moisture content on the two slopes under a rainfall density of 1.0 mm/min, and the moisture distribution process for the period following the end of the rainfall application. It is apparent from the figure that for the first rainfall, the rising point of the soil's water content curve is significantly different in the sand-covered slope compared to that in the loess slope at different depths. At soil depths of 3 cm, 6 cm, 9 cm, 14 cm and 22 cm, the growth times of the soil moisture content curves with duration of rainfall were 1 min, 1 min, 7 min, 13 min and 33 min, respectively. The change curves of soil moisture content at different depths of the loess slope showed an increasing trend, with growth times of 1 min, 2 min, 16 min, 22 min and 91 min, respectively.

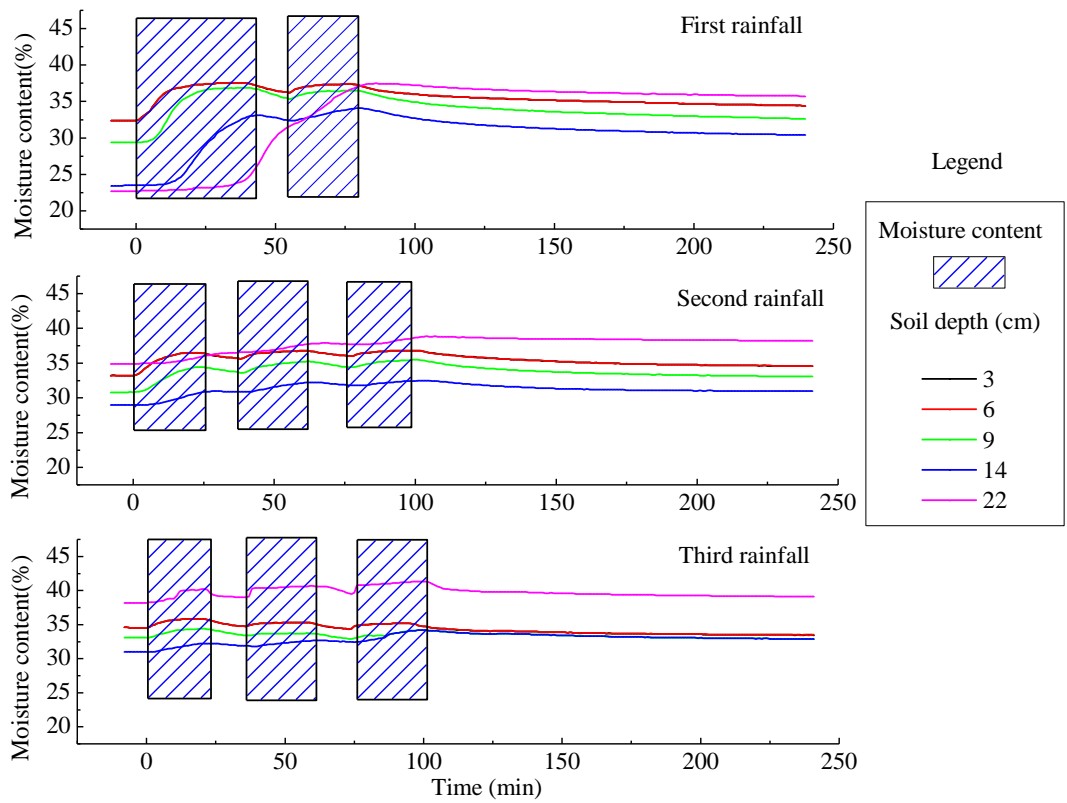

**Figure 5.** Dynamic change process of soil water content of the sand-covered slope at a rainfall density of 1.0 mm/min.

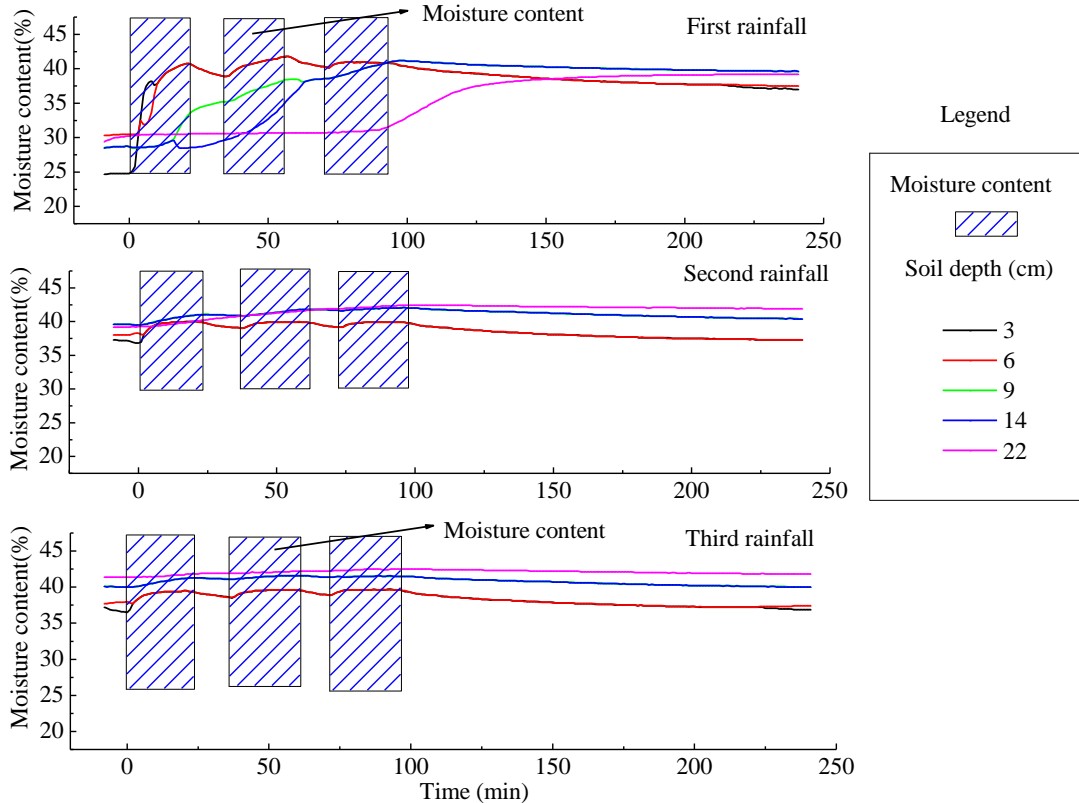

**Figure 6.** Dynamic change process of soil water content of the loess slope at a rainfall density of 1.0 mm/min.

### 3.2.2. Spatial and Temporal Distribution Characteristics of Soil Moisture on the Slope Surface under a Rainfall Density of 1.5 mm/min

Figures 7 and 8 display dynamic change process of the soil moisture content at different depths, as well as the water distribution process, for the two different kinds of slopes, associated with a rainfall density of 1.5 mm/min, and it was observed that for the first rainfall of the sand-covered slope, at depths of 3 cm, 6 cm, 9 cm, 14 cm and 22 cm, initial growth times of the moisture content curves with the duration of rainfall were 1 min, 1 min, 7 min, 18 min and 90 min, respectively. The initial growth time of the water content curves of the loess slope were 1 min, 4 min, 9 min, 26 min and 93 min, respectively. It can also be seen that soil moisture of the sand-covered slope may reach deep soil earlier within the same duration of rainfall. Here, the infiltration water first resulted in an increase in soil moisture content in shallow soil, with the soil moisture content at a soil depth of 6 cm being the first to approach 40%. Near the end of the rainfall application, the soil moisture content at depths of 3, 6, 9 and 14 cm of the sand-covered slope were basically the same. In addition, from the water distribution process, following the second rainfall, the soil moisture content at the depth of 22 cm remained the highest. In a subsequent rainfall experiment, the soil moisture content at a depth of 22 cm remained the highest, and the remaining water content curves showed a decreasing trend with increasing soil depth. Finally, the soil moisture content at different depths of the sand-covered slope were basically the same, while the soil moisture content at different depths of the loess slope were different.

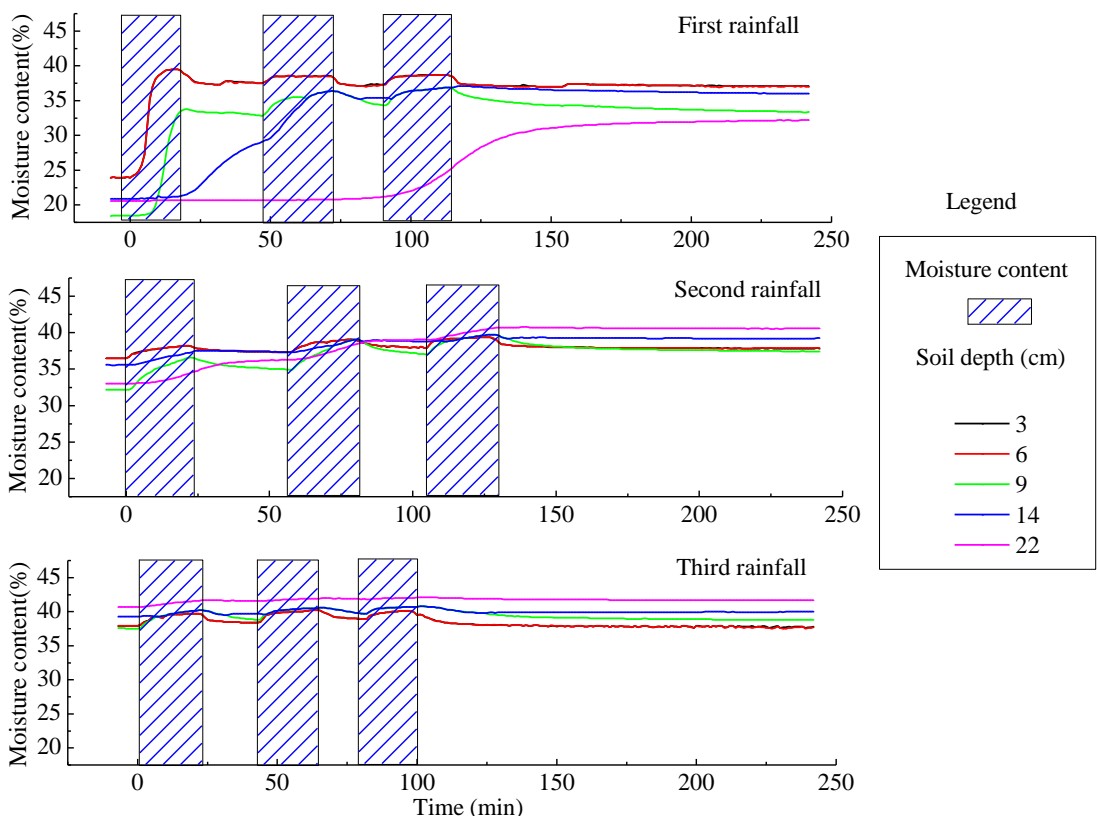

**Figure 7.** Dynamic change process of soil water content of the sand-covered slope at a rainfall density of 1.5 mm/min.

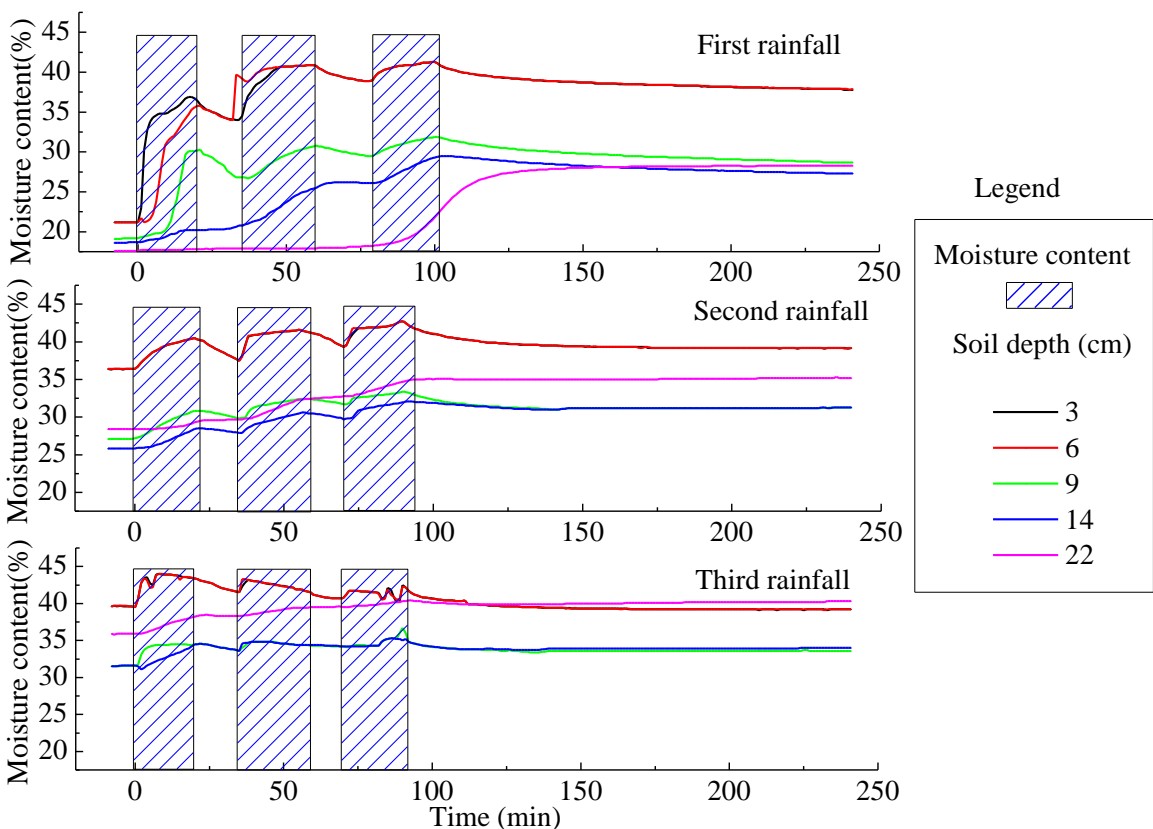

**Figure 8.** Dynamic change process of soil water content of the loess slope at a rainfall density of 1.5 mm/min.

### 3.2.3. Spatial and Temporal Distribution Characteristics of Soil Moisture on the slope Surface under a Rainfall Density of 2.0 mm/min

Figures 9 and 10 show the dynamic change and distribution process of soil moisture content on the two slopes under a rainfall density of 2.0 mm/min. It can be seen from the map that in the first rainfall, the soil moisture content at depths of 3 cm, 6 cm, 9 cm, 14 cm and 22 cm increased after 1 min, 2 min, 6 min, 16 min and 40 min, respectively. The growth times of the water content curves of the loess slope were 1 min, 1 min, 8 min, 24 min and 86 min, respectively. This observation indicates that soil moisture of the sand-covered slope may reach deep soil earlier within the same rainfall duration compared to of the loess slope. The infiltration water first resulted in an increase in soil moisture content in the shallow soil, with the soil moisture content at soil depths of 3 and 6 cm being the first to approach 43%. In the subsequent duration of rainfall, the soil moisture content remained the highest at a depth of 22 cm.

### 3.2.4. Spatial and Temporal Distribution Characteristics of Soil Moisture on the Slope Surface under a Rainfall Density of 2.5 mm/min

Figures 11 and 12 show the dynamic change of soil moisture content and water distribution process of 2.5 mm/min rainfall density on both slopes. Here, during the first rainfall of the sand-covered slope, the soil moisture content of 3 cm, 6 cm, 9 cm, 14 cm and 22 cm soil depths increased at the first minute, 1 min, 3 min, 8 min and 27 min, respectively. The growth time of soil moisture content curve of the loess slope were 2 min, 5 min, 11 min, 24 min and 60 min, respectively. The above phenomena suggest that soil moisture may reach deeper soil earlier in the same duration of rainfall. The infiltration water first increased the soil moisture content in shallow soil, and the soil moisture content at 6 cm soil depth was the first to approach 38%. Following the first rainfall, the soil moisture content at the depth of 22 cm remained the highest. In the subsequent two rainfall experiments,

the soil moisture content at 22 cm depth remained the highest, and the remaining water content curves showed a decreasing trend with increased soil depth.

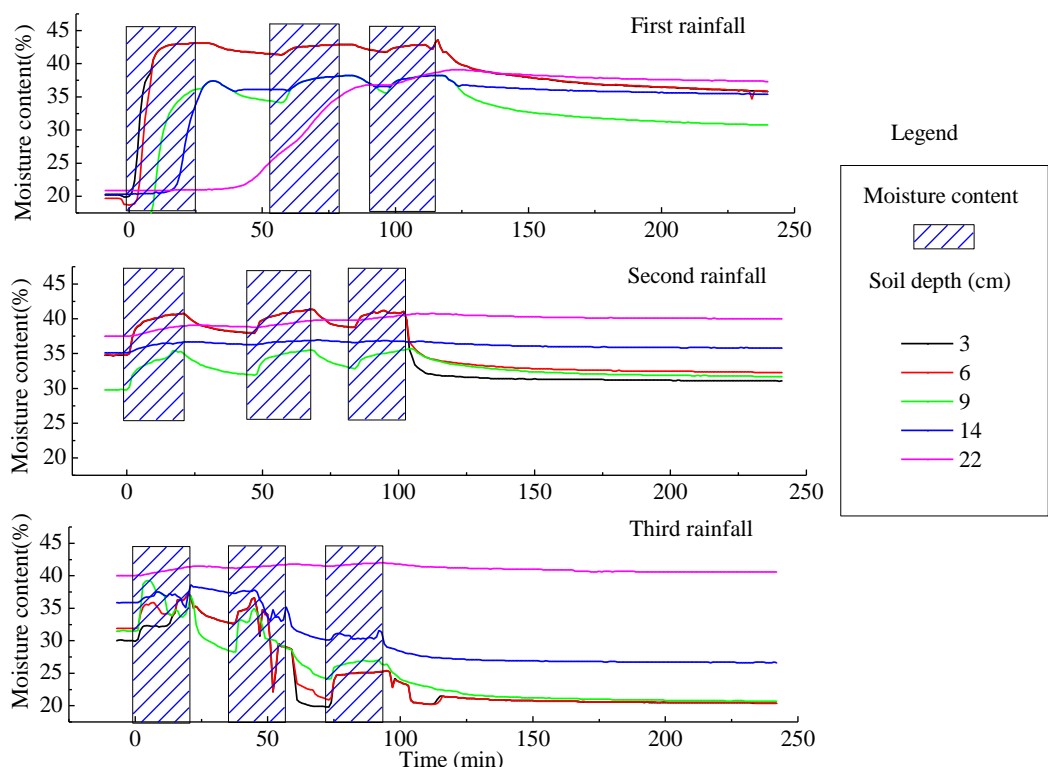

**Figure 9.** Dynamic change process of soil water content of the sand-covered slope at a rainfall density of 2.0 mm/min.

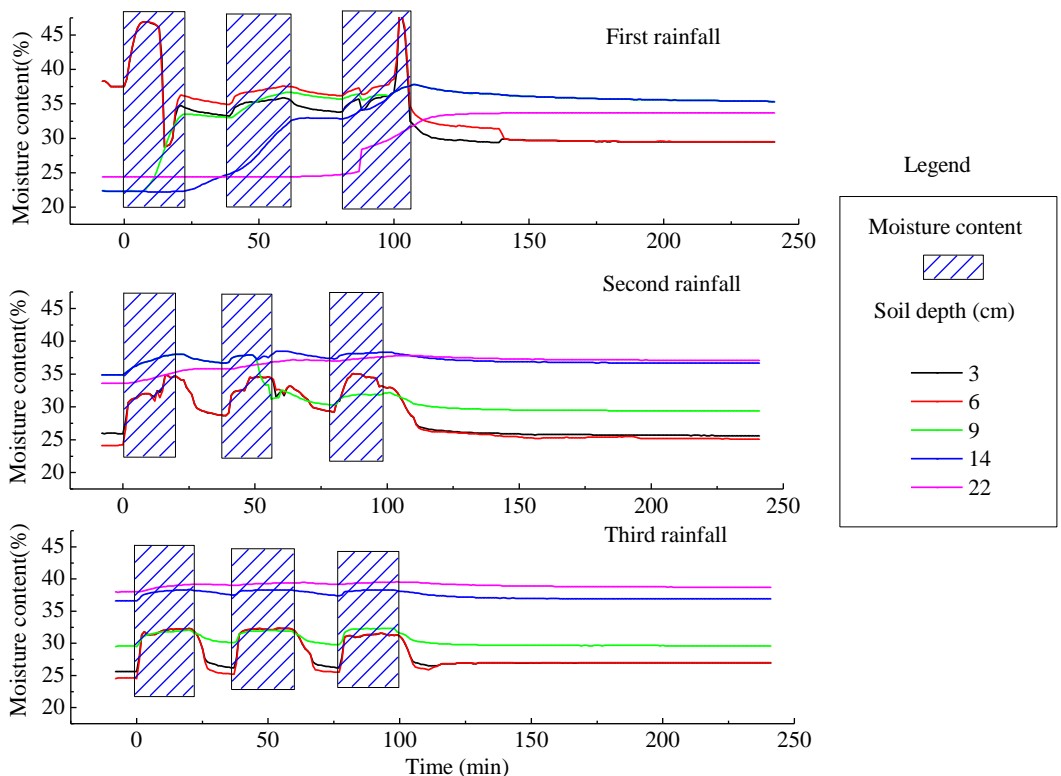

**Figure 10.** Dynamic change process of soil water content of the loess slope in 2.0 mm/min rain.

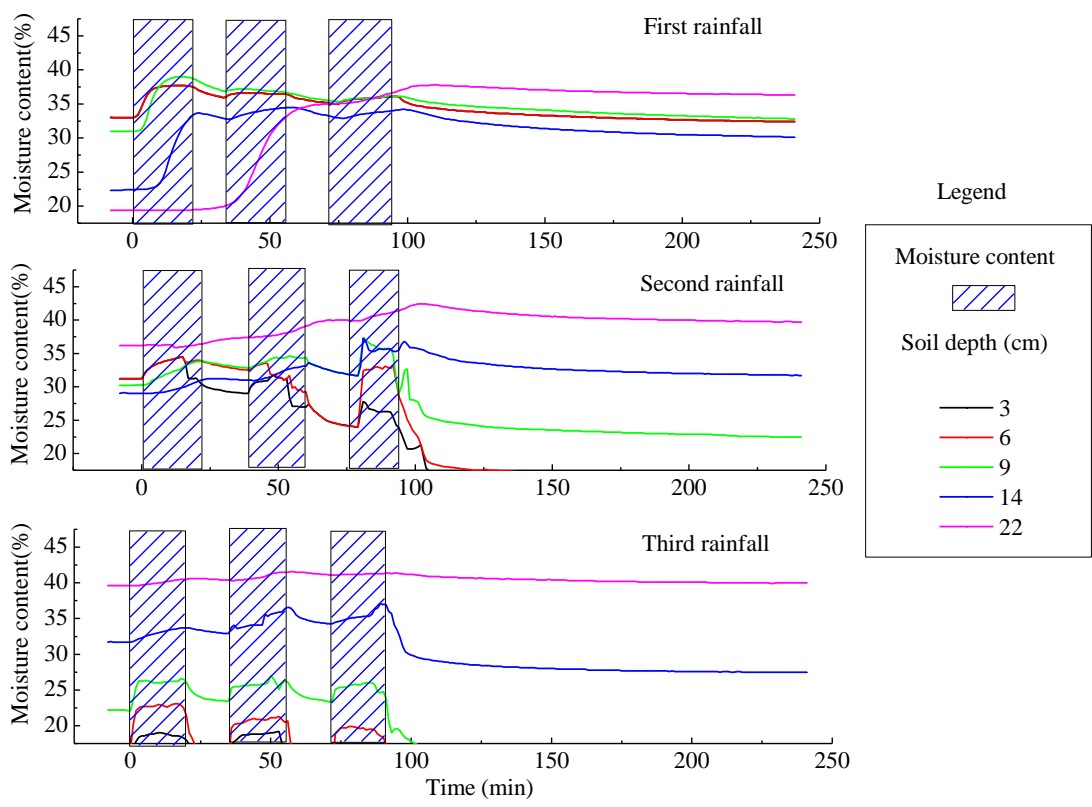

**Figure 11.** Dynamic change process of soil water content of the sand-covered slope in 2.5 mm/min rain.

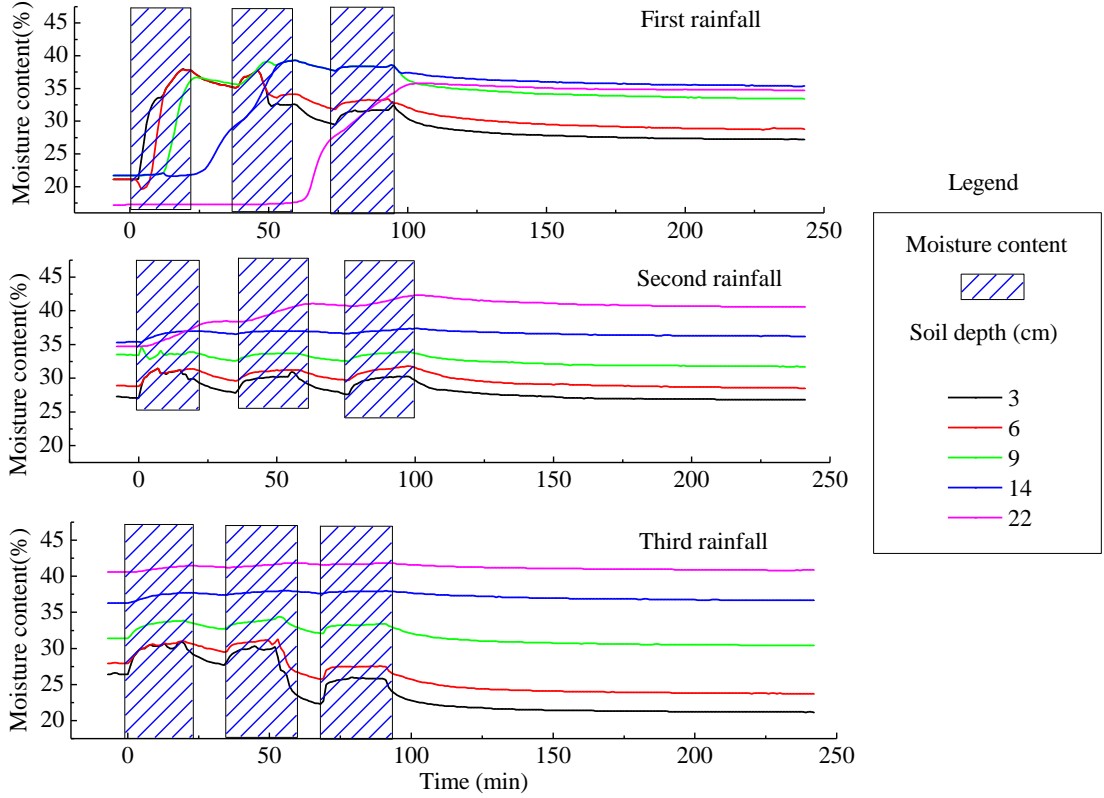

**Figure 12.** Dynamic change process of soil water content of the loess slope in 2.5 mm/min rain.

Overall, as shown in Table 3, the rising time of soil moisture content curve at different depths of the sand-covered slope was measured. Here, aside from the rainfall density of 1.0 mm/min, the following trends were observed: the greater the rainfall density, the earlier the rising time of soil moisture content curve at different depths, and the initial runoff time was advanced with increased rainfall density. Due to the rainfall density of 1.0 mm/min, the initial runoff time is later, about 15 min, and the infiltration before runoff is the largest across the four rainfall densities, so the time of water reaching deep soil is earlier. With greater rainfall density, the water supply conditions are sufficient, and the vertical movement of soil moisture of the sand-covered slope is more favorable, resulting in the early rise time of the moisture content curve.

**Table 3.** The rising time of soil water content curves at different depths on sand-covered slopes.

| Rainfall Density (mm/min) | Soil Depth | | | | | Initial Runoff Time (min) |
|---|---|---|---|---|---|---|
| | 3 cm | 6 cm | 9 cm | 14 cm | 22 cm | |
| 1.0 | 1 | 1 | 7 | 13 | 33 | 15 |
| 1.5 | 1 | 1 | 7 | 18 | 90 | 6 |
| 2.0 | 1 | 2 | 6 | 16 | 40 | 3 |
| 2.5 | 1 | 1 | 3 | 8 | 27 | 2 |

Statistics for the rising time of soil moisture curve at different depths of the loess slope are displayed in Table 4. It is clear that the rising time of soil moisture curve at deeper depths is advanced with increased rainfall density, and the rising time of the soil moisture curve at other depths is slightly different.

**Table 4.** The rising time of soil water content curves at different depths on loess slopes.

| Rainfall Density (mm/min) | Soil Depth | | | | | Initial Runoff Time (min) |
|---|---|---|---|---|---|---|
| | 3 cm | 6 cm | 9 cm | 14 cm | 22 cm | |
| 1.0 | 1 | 2 | 16 | 22 | 91 | 3 |
| 1.5 | 1 | 4 | 9 | 26 | 93 | 1 |
| 2.0 | 1 | 1 | 8 | 24 | 86 | 1 |
| 2.5 | 2 | 5 | 11 | 24 | 60 | 1 |

*3.3. Soil Moisture Parameter Change Process of the Slopes*

3.3.1. Slope Runoff Velocity

Figures 13 and 14 represent the curves of velocity over time on the sandy slope and loess slope, respectively, under three rainfall density levels. Under different rainfall densities, the variation ranges of runoff velocity of the sand-covered slope and loess slope were 0.014–0.167 m/s and 0.046–0.136 m/s, respectively. Of the sand-covered slope, the runoff velocity generally increased with increasing rainfall density, but at rainfall densities of 2.0 and 2.5 mm/min, the overall difference in runoff velocity was not significant.

3.3.2. Slope Runoff Shear Stress

Figures 15 and 16 show the variation curves of runoff shear stress with duration of rainfall of the sand-covered slope and loess slope, respectively. In a word, the runoff shear stress increased with extended rainfall, and then stabilized after a certain rainfall duration was reached. Significant differences were observed between the three rainfall applications on the same slope. The trend of runoff shear stress with the three rainfall applications was as follows: third rainfall > second rainfall > first rainfall. With increasing rainfall density, there was also an increase in runoff shear stress, indicating that there was an increase in the ability of water to erode soil particles. The average values of runoff shear stress for the three applications of intermittent rainfall on the slopes at different rainfall densities are outlined in Table 5. These results indicate that the runoff shear stress of the loess slope is greater than that of the sand-covered slope under different rainfall densities.

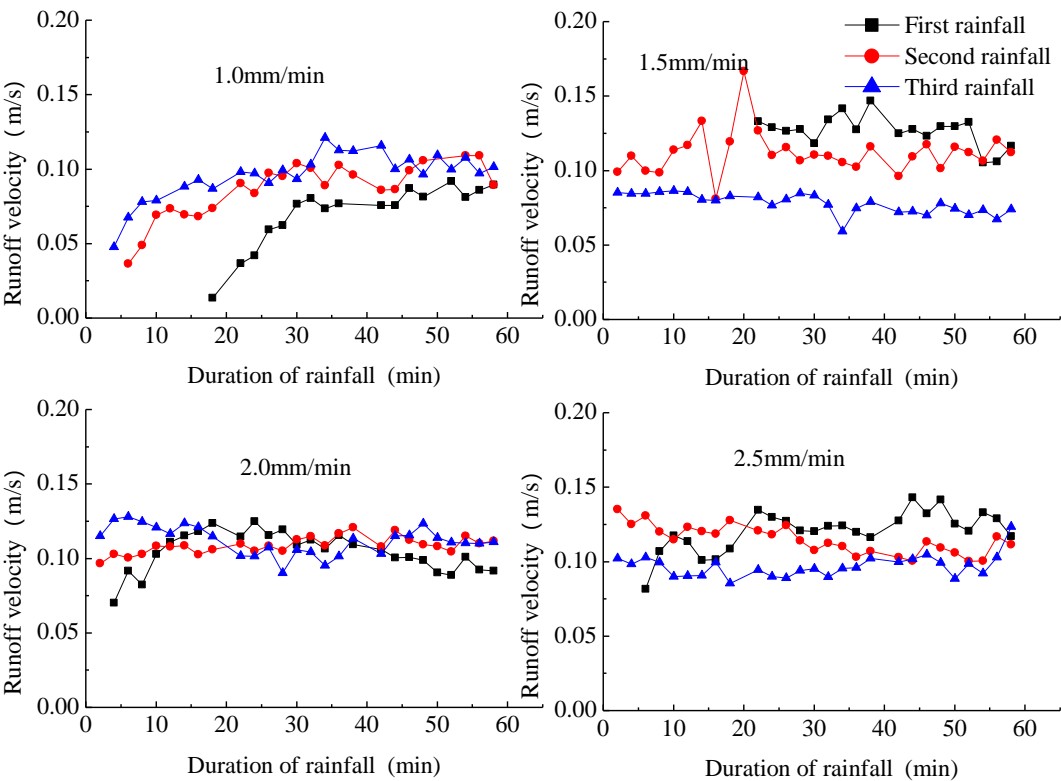

**Figure 13.** Variation curve of runoff velocity with time of the sand-covered slope at different rainfall densities.

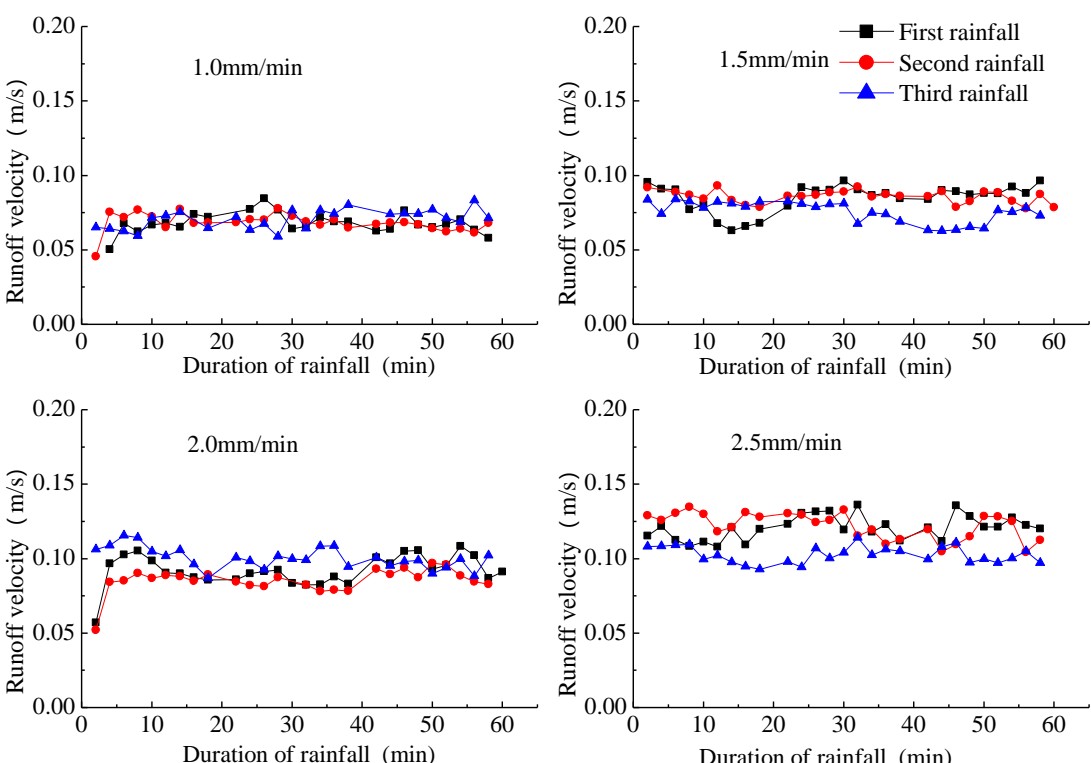

**Figure 14.** Variation curve of runoff velocity with time of the loess slope at different rainfall densities.

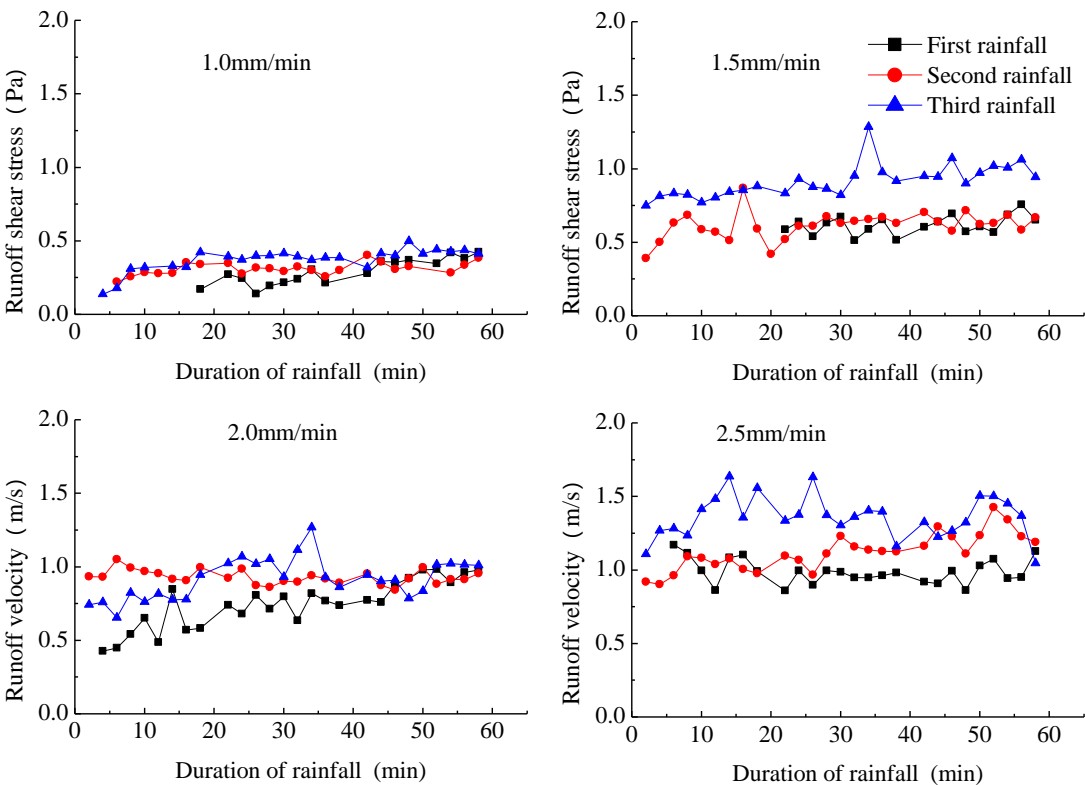

**Figure 15.** Variation curve of runoff shear stress with time of the sand-covered slope at different rainfall densities.

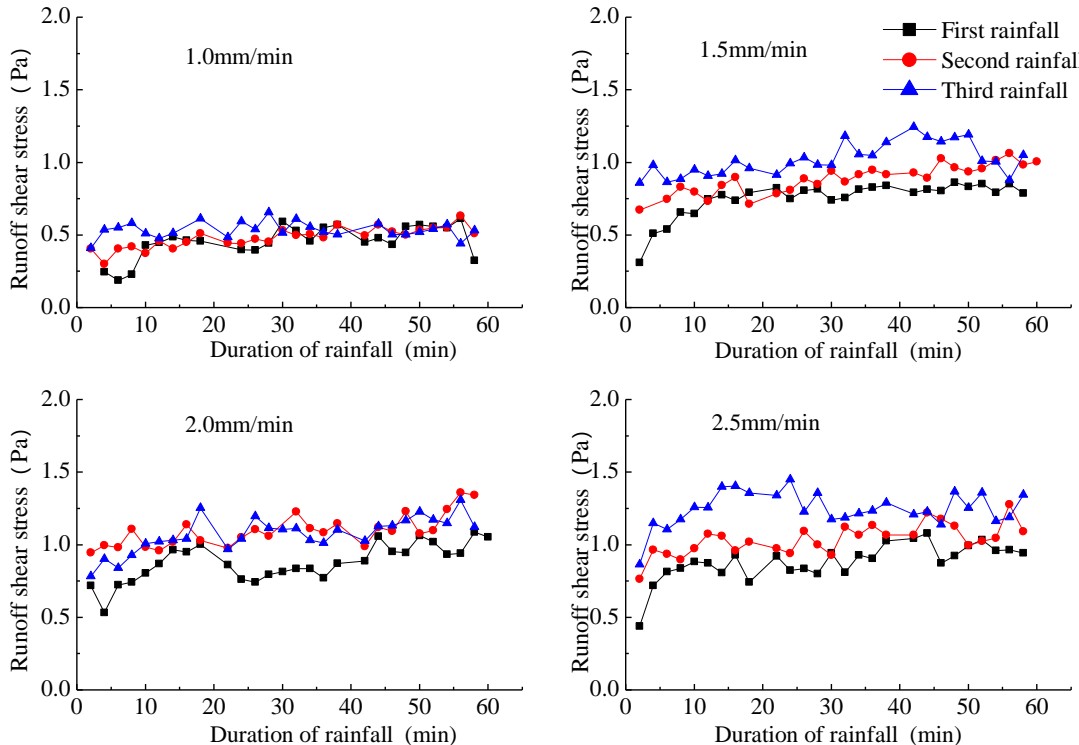

**Figure 16.** Variation curve of runoff shear stress with time of the loess slope at different rainfall densities.

**Table 5.** Average runoff shear stress of downslope with different rainfall densities and rainfall applications.

| Sand-Covered Slope (Pa) | | | | | Slope Land (Pa) | | | | |
|---|---|---|---|---|---|---|---|---|---|
| Rainfall density (mm/min) | 1.0 | 1.5 | 2.0 | 2.5 | Rainfall density (mm/min) | 1.0 | 1.5 | 2.0 | 2.5 |
| First rainfall | 0.29 | 0.62 | 0.75 | 0.99 | First rainfall | 0.46 | 0.75 | 0.88 | 0.88 |
| Second rainfall | 0.31 | 0.62 | 0.93 | 1.12 | Second rainfall | 0.48 | 0.89 | 1.10 | 1.04 |
| Third rainfall | 0.37 | 0.92 | 0.92 | 1.36 | Third rainfall | 0.54 | 1.02 | 1.07 | 1.25 |

The rainfall density under different rainfall applications was linearly fitted with the corresponding average runoff shear stress of the sand-covered slope, and the results are shown in Figure 17. The results suggest a strong linear relationship between rainfall density and the average runoff shear stress on the slope. When the first rainfall occurred of the sand-covered slope, the average runoff shear stress increased by 0.223 Pa for every 0.5 mm/min increase in rainfall density. At a rainfall density of 0.5 mm/min, the average runoff shear stress increased by 0.274 Pa. At a rainfall density of 0.5 mm/min during the third rainfall application of the sand-covered slope, the average runoff shear stress increased by 0.297 Pa.

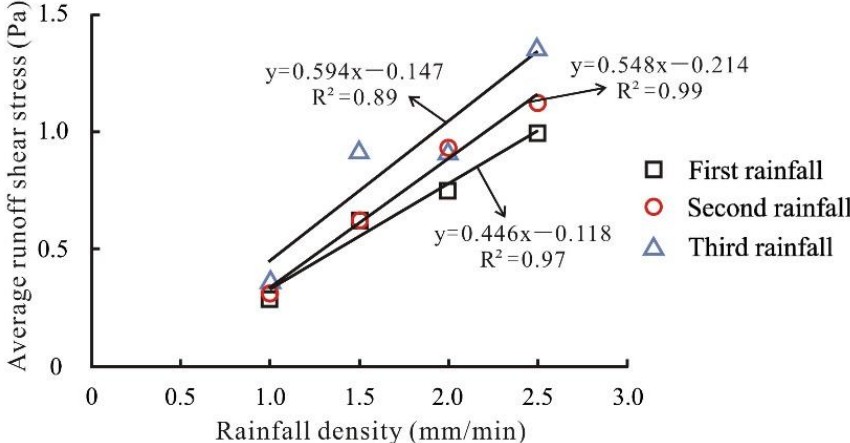

**Figure 17.** Linear fitting relationship between average runoff shear stress and rainfall density of the sand-covered slope.

Similarly, the rainfall density under different rainfall applications was linearly fitted with the corresponding average runoff shear stress of the loess slope, as shown in Figure 18. Like the sand-covered slope, it can be seen from the figure that there is an obvious linear relationship between rainfall density and average runoff shear stress of the loess slope. In this case, at a rainfall density of 0.5 mm/min in the first rainfall of the loess slope, the average runoff shear stress increased by 0.139 Pa. When the rainfall density increased by 0.5 mm/min, the average runoff shear stress increased by 0.189 Pa. At a rainfall density of 0.5 mm/min during the third rainfall application of the loess slope, the average runoff shear stress increased by 0.218 Pa.

### 3.3.3. Slope Runoff Power

Figures 19 and 20 show the variation curves of runoff power with time of the sand-covered slope and the loess slope, respectively. Here, the variation in runoff power with time exhibited a slow increase at first and then stabilized, but there were clear differences in the variation of runoff power of the two slopes at different rainfall densities. The average runoff power on the slope was determined, and the results are shown in Table 6.

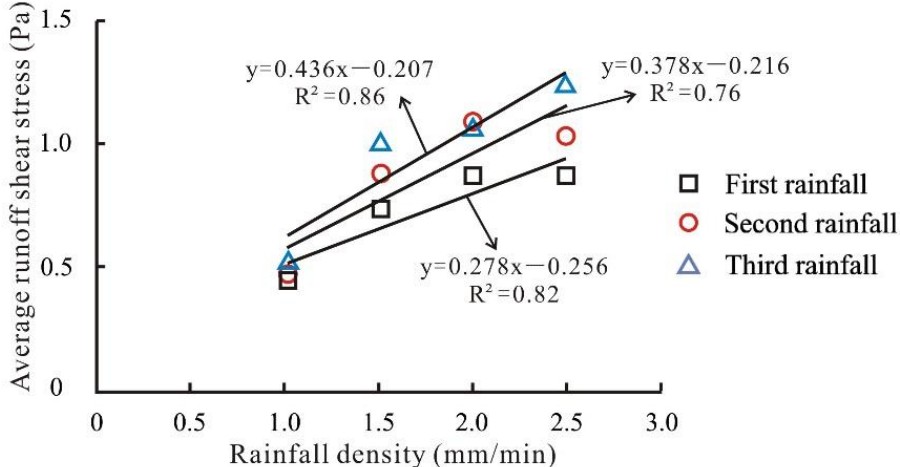

**Figure 18.** Linear fitting relationship between average runoff shear stress and rainfall density of the loess slope.

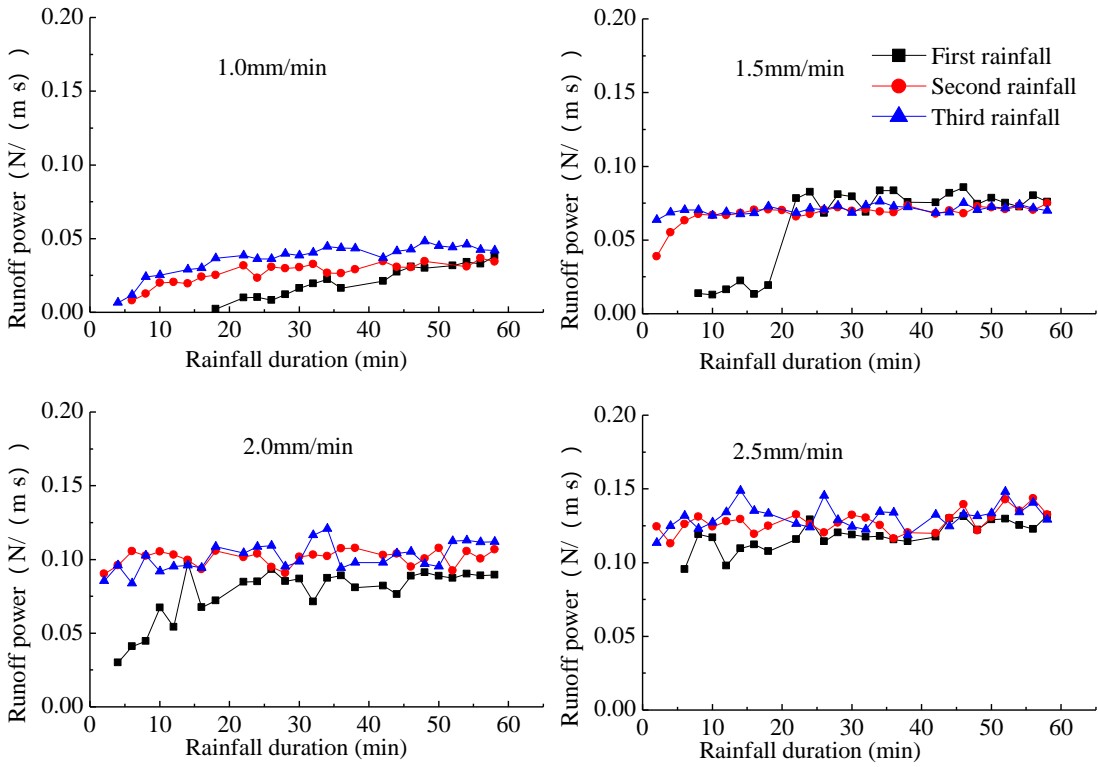

**Figure 19.** Variation curve of runoff power with time of the sand-covered slope.

**Table 6.** Average downslope runoff power with different rainfall density and times.

| Sand-Covered Slope (N/(m·s)) | | | | | Loess Slope (N/(m·s)) | | | | |
| --- | --- | --- | --- | --- | --- | --- | --- | --- | --- |
| Rainfall density (mm/min) | 1.0 | 1.5 | 2.0 | 2.5 | Rainfall density (mm/min) | 1.0 | 1.5 | 2.0 | 2.5 |
| First rainfall | 0.022 | 0.063 | 0.078 | 0.119 | First rainfall | 0.031 | 0.064 | 0.081 | 0.107 |
| Second rainfall | 0.027 | 0.068 | 0.101 | 0.128 | Second rainfall | 0.033 | 0.076 | 0.095 | 0.126 |
| Third rainfall | 0.037 | 0.071 | 0.102 | 0.131 | Third rainfall | 0.038 | 0.077 | 0.107 | 0.128 |

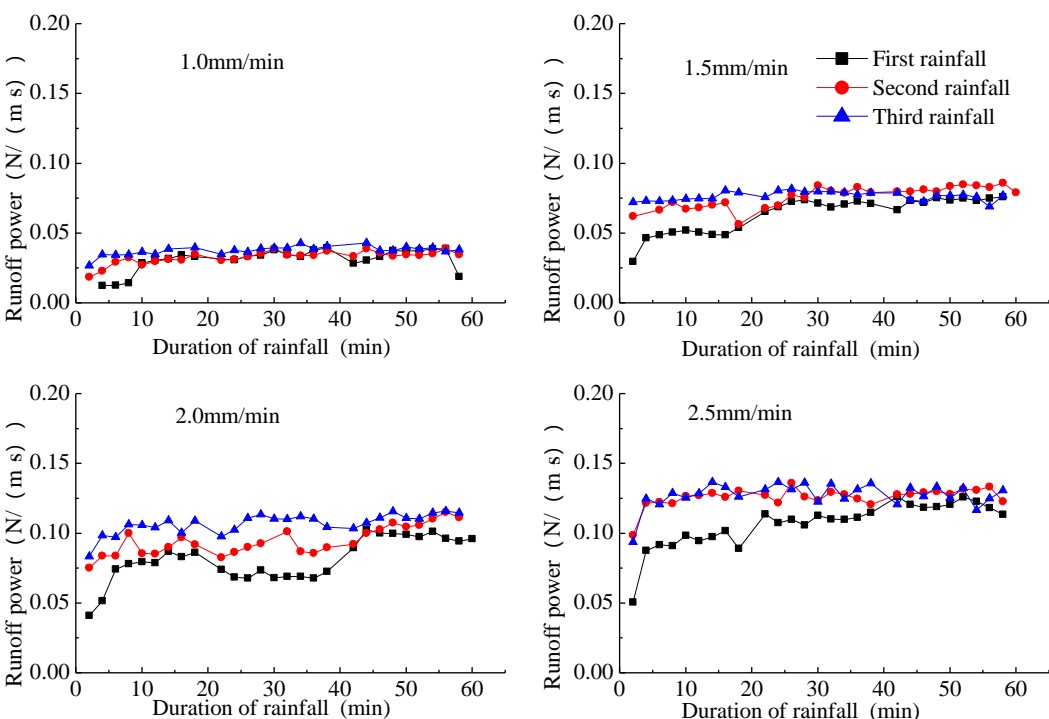

**Figure 20.** Variation curve of runoff power with time of the loess slope.

Clearly, there are obvious differences among the three rainfall applications occurring on the same slope, and the runoff power relationship between the three rainfall applications is: the third rainfall > the second rainfall > the first rainfall. The runoff power increased with increasing rainfall density, indicating that the ability of water to transport soil particles had an increasing trend. Under different rainfall densities, the runoff power of the loess slope was greater than that on the sandy slope.

Linear regression was used to fit the rainfall density under different fields with the corresponding average runoff power of the sand-covered slope, and the results are shown in Figure 21. It can be seen that there is also a good linear relationship between rainfall density and the average runoff power of the sand-covered slope. When the first rainfall occurred of the sand-covered slope, the average runoff power increased by 0.031 N/(m·s) for every 0.5 mm/min increase in rainfall density. For the second rainfall, at a rainfall density of 0.5 mm/min of the loess slope, the average runoff power increased by 0.034 N/(m·s). For the third rainfall, at a rainfall density of 0.5 mm/min of the loess slope, the average runoff power increased by 0.031 N/(m·s).

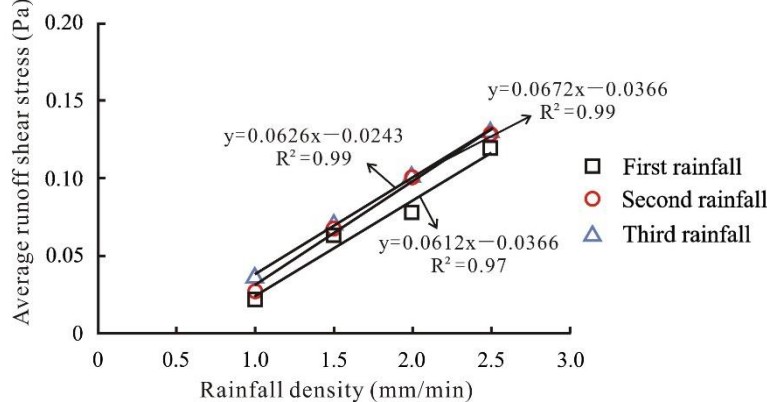

**Figure 21.** Linear fitting relationship between average runoff power and rainfall density of the sand-covered slope.

Using the above method, the relationship between rainfall density and average runoff power of the loess slope was determined for different rainfall applications. As shown in Figure 22, there is also a clear linear relationship between rainfall density and average runoff power of the loess slope. When the first rainfall occurred of the loess slope, the average runoff power increased by 0.025 N/(m·s) for every 0.5 mm/min increase in rainfall density. At a rainfall density of 0.5 mm/min, the average runoff power increased by 0.030 N/(m·s). For the third rainfall, at a rainfall density of 0.5 mm/min of the loess slope, the average runoff power increased by 0.03 N/(m·s).

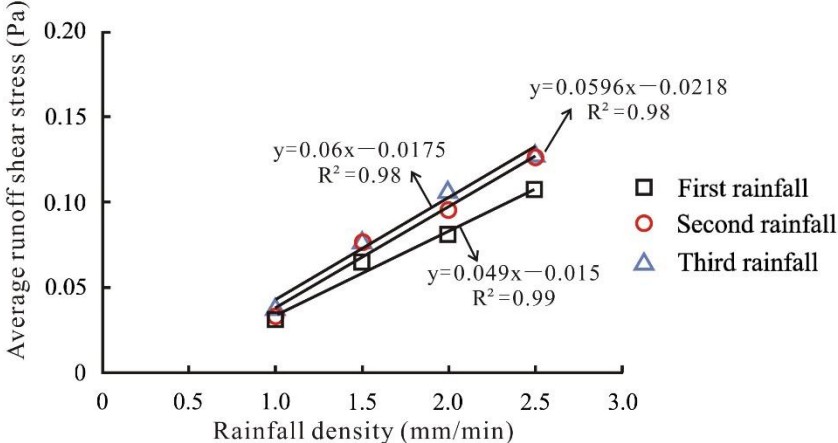

**Figure 22.** Linear fitting relationship between average runoff power and rainfall density of the loess slope.

## 4. Discussion

### 4.1. Sand-Covered Slope

In this study, the infiltration velocity of the soil was observed to decrease with increased rainfall. On one hand, the soil moisture content increases with increasing rainfall duration, which leads to a decrease in soil infiltration capacity [33,34]. On the other hand, the soil surface may gradually form a crust during rainfall, further preventing the infiltration of soil moisture [35–37]. There are two main differences in the infiltration law between the sand-covered and loess slopes, namely, the initial infiltration velocity and the change rate of the infiltration velocity. During the first 20 min of rainfall, the sand-covered slope showed a greater soil infiltration velocity than that of the loess slope. However, it was observed that the rate of change of the infiltration velocity of the loess slope was greater than that of the sand-covered slope, so the time required to achieve stable infiltration of the loess slope is likely shorter. This is because the porosity of the sand layer on the surface is high, and the water is more likely to leak in the vertical direction. In addition, because the sand layer stores some rainfall, the amount of rain that seeps into the surface of the loess soil layer is lower than the amount of the actual rainfall, so the rainfall density at the initial stage does not exceed the infiltration capacity of the soil. Therefore, the infiltration velocity of the sand-covered slope is greater than that of the loess slope during this period [13,38,39].

With the same duration of rainfall, the soil moisture reaches the deep soil earlier of the sand-covered slope than of the loess slope, with greater infiltration. This is likely because the sand layer of the sand-covered slope surface weakens the energy and the size of raindrops. In addition, soil crust cannot be generated under the sand layer conditions, resulting in increased soil infiltration capacity. The results of this study indicate that before steady infiltration is achieved, the water infiltration velocity of the sand-covered slope will always be greater than that of the loess slope, so the infiltration amount will also be greater. In the process of infiltration, the soil moisture content of the shallow soil increases first. By comparing the runoff time, it was found that the soil moisture content at a depth of 6 cm reached a certain value of the sand-covered slope, at which point runoff began, indicating that the shallow soil was nearly saturated. After the runoff generation time,

the water can be divided into two categories. One part generates surface runoff, while the other part is used in the process of infiltration. At this point, the infiltration amount of the sand-covered slope was still greater than that of the loess slope, resulting in the increase in water content in the deep soil occurring earlier of the sand-covered slope than of the loess slope. Following rainfall, the soil moisture content at different depths was generally consistent of the sand-covered slope. In addition, it was found from the water distribution process that the soil moisture content at a depth of 22 cm remained the highest after rainfall. In the two subsequent rainfall experiments, the soil moisture content at a depth of 22 cm remained the highest, and the remaining water content curves showed a decreasing trend with increasing soil depth. Finally, the soil moisture content at different depths were basically the same between the slopes.

### 4.2. Soil Moisture Changes

Compared to the loess slope, the runoff velocity of the sand-covered slope fluctuated significantly with time, mainly as a result of the characteristics of sediment production. Due to the special physical properties of sand, the sand–soil dual structure becomes fragile, and the slope is therefore more prone to erosion and collapse under the action of hydraulic effects and self-gravity [10–13]. In this way, flow channels will be blocked at a given point, and subsequently washed away; therefore, runoff velocity shows great fluctuation. Of the loess slope, the runoff velocity did not fluctuate greatly with the duration of rainfall, indicating that the flow characteristics were stable throughout the entire rainfall process. The runoff velocity increased with increasing rainfall density, which is likely because increased rainfall density leads to increased water depth and hydraulic radius in the rain-affected area, whereby the resistance coefficient is decreased, and the runoff velocity increases with increasing rainfall density [40,41].

The water infiltration of the sand-covered slope was greater than that of the loess slope. Therefore, the total runoff of the sand-covered slope was observed to be less than that of the loess slope per time unit, resulting in the thin-layer flow of the sand-covered slope having a shallow water depth. Similarly, the runoff shear stress was lower than that of the loess slope. Under the same rainfall density, the runoff shear stress of the loess slope was greater than that of the sand-covered slope, but the runoff shear stress of the sand-covered slope increased more rapidly with increasing rainfall density, that is, the slope in the linear relationship was steeper. At a rainfall density of 0.5 mm/min, the runoff shear stress of the sand-covered slope increased at a rate of about 1.5 times that of the loess slope.

The runoff power [42,43] also suggests that the runoff shear stress of the sand-covered slope was greater than that of the loess slope. The equation for estimating runoff power highlights its close relationship with runoff shear stress and velocity. In this study, the runoff shear stress on the sediment-covered slope was greater than that of the loess slope, and there was no significant difference in the average velocity of the two slopes. Therefore, it was determined that the runoff power of the sand-covered slope was lower than that of the loess slope. Under the same rainfall density, the runoff power of the loess slope was generally greater than that of the sand-covered slope. However, the runoff power of the sand-covered slope increased slightly with increasing rainfall density. At a rainfall density of 0.5 mm/min, the increase of runoff power of the sand-covered slope was about 1.13 times that of the loess slope.

### 5. Conclusions

On the basis of our observation of the processes of infiltration, flow generation, water flow characteristics, and the spatial distribution erosion during a designed rainfall test, and analyzing the infiltration, flow generation characteristics, water content change characteristics, soil moisture parameter change characteristics, and changes in the spatial patterns of erosion and sediment yield, this study draws the following conclusions:

Under different rainfall densities, the initial runoff generation time of the sediment-covered slope was 1~12 min longer than that of the loess slope; the initial soil infiltration

rate of the sediment-covered slope was about 1.23 times that of loess slope; and the time taken to achieve stable infiltration of the loess slope was shorter than that of the sediment-covered slope.

Under different rainfall intensities, the rising time of the water content curve of the sand-covered slope was shorter than that of the loess slope. Within the same duration of rainfall, the vertical infiltration performance of soil water of the sand-covered slope was higher than that of the loess slope.

When the rainfall intensity on the slope increased by 0.5 mm/min, the increase in the value of runoff shear force on the sediment-covered slope was about 1.5 times that of the loess slope, and the runoff power was about 1.13 times that of the loess slope.

**Author Contributions:** F.W. and G.X. conceived the main idea of the paper. Z.L., P.L., T.W., J.Z., J.W. and Y.C. designed and performed the experiment. F.W. wrote the manuscript and all authors contributed to improving the paper. All authors have read and agreed to the published version of the manuscript.

**Funding:** This research was supported by the National Natural Science Foundation of China (U2040208), the National Natural Science Foundation of China (52009104, 42167007, 42007069), the Science and Technology Project of Department of Transport of Shaanxi Province (Grant NO. 2015-11K).

**Institutional Review Board Statement:** Not applicable.

**Informed Consent Statement:** Not applicable.

**Data Availability Statement:** Not applicable.

**Acknowledgments:** We thank the reviewers for their useful comments and suggestions. In addition, we would like to thank Shannon Elliot at Michigan State University for his assistance with English language and grammatical editing.

**Conflicts of Interest:** The authors state that they have no known competing financial interest or personal relationships that could affect the work described in this article.

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
