# Peer review of "Study on Infiltration and Soil Moisture Characteristics of a Sand-Covered Slope"

_water, doi:10.3390/w14071043_

Round 1

Reviewer 1 Report

Title can be modified. Hydrodynamics can be replaced with Soil Moisture ...

Page 2: Wu et al. cited but missing in the Reference section

Test material: Line 5 - you have mentioned 0.30 m high, but in the Figure it is 0.6 m.

Page 4: Check the English. 'is was determined' ???

Page 5: Lines 6, 7 - collect instead of connect.

Captions of Figures 3-1, 3-2: Delete 'variation process of'

Page 8: Line 9: not clear... Improve the readability. 'were 1 min... , respectively.'

Figure 3-6: Duration of raining repeated already mentioned in the Legend.

Tables 3-2 and 3-3 can be merged into a single Table.

What do you mean by hydrodynamic properties? Essentially you have studied the soil moisture characteristics and the runoff intensity. Simple word usage may improve the clarity of the manuscript. Please avoid the usage of words such as dynamics etc.

Rainfall field? do you mean intensity?

The MS is within the scope of the journal. It is a well planned study and has been written well. However, please consult a native English speaking person to further improve the MS.

Author Response

Dear editor and reviewers. Please see the attachment.

Reviewer 2 Report

I have some suggestions and corrections that would probably elevate the paper:

General comments

  1. The innovative points of your research must be given at the Introduction section. It is already known that infiltration rate decreases with time, so you have to give the novelty of your study.
  2. Researchers that investigate the phenomenon of infiltration, which strongly affects surface runoff, take into consideration some crucial hydraulic parameters of the soil, which are related to the infiltration process. Some of them are: Soil sorptivity, difussivity, hydraulic conductivity, soil water retention curve, initial soil moisture, soil moisture at saturation, capacity, etc. Did you connect your research with the hydraulic parameters of the soils under research?
  3. Worldwide literature is missing and must be added into the Introduction section. I suggest some innovative researches, below:
    1. Schmid, B.H. On overland flow modelling: Can rainfall excess be treated as independent of flow depth? (1989) Journal of Hydrology, 107 (1-4), pp. 1-8. doi: 10.1016/0022-1694(89)90045-0
    2. Angelaki, A., Sakellariou-Makrantonaki, M., Tzimopoulos, C. Laboratory experiments and estimation of cumulative infiltration and sorptivity (2004) Water, Air, and Soil Pollution: Focus, 4 (4-5), pp. 241-251. doi: 10.1023/B:WAFO.0000044802.21695.25
  • Mertens, J., Stenger, R., Barkle, G.F. Multiobjective inverse modeling for soil parameter estimation and model verification (2006) Vadose Zone Journal, 5 (3), pp. 917-933. http://vzj.scijournals.org/cgi/reprint/5/3/917 doi: 10.2136/vzj2005.0117
  1. Sakellariou-Makrantonaki M., Angelaki A., Evangelides C., Bota V., Tsianou E., Floros N., Experimental Determination of Hydraulic Conductivity at Unsaturated Soil Column, Procedia Engineering, Volume 162, (2016) Pages 83-90,https://doi.org/10.1016/j.proeng.2016.11.019. https://www.sciencedirect.com/science/article/pii/S1877705816333240
  2. Mishra, S.K., Chaudhary, A., Shrestha, R.K., Pandey, A., Lal, M. Experimental verification of the effect of slope and land use on scs runoff curve number (2014) Water Resources Management, 28 (11), pp. 3407-3416. www.wkap.nl/journalhome.htm/0920-4741 doi: 10.1007/s11269-014-0582-6
  3. Muntohar, A.S., Liao, H.-J. Factors affecting rain infiltration on a slope using Green-Ampt model (Open Access) (2019) Journal of Physical Science, 30 (3), pp. 71-86. http://jps.usm.my/wp-content/uploads/2019/11/JPS-303_Art5-71-86.pdf doi: 10.21315/jps2019.30.3.5
  • Pandey, P.K., Pandey, V. Estimation of infiltration rate from readily available soil properties (RASPs) in fallow cultivated land (2019) Sustainable Water Resources Management, 5 (2), pp. 921-934. springer.com/journal/40899 doi: 10.1007/s40899-018-0268-y
  • Guellouz, L., Askri, B., Jaffré, J., Bouhlila, R. Estimation of the soil hydraulic properties from field data by solving an inverse problem (2020) Scientific Reports, 10 (1), art. no. 9359. nature.com/srep/index.html doi: 10.1038/s41598-020-66282-5
  1. Angelaki, A., Sihag, P., Sakellariou-Makrantonaki, M., Tzimopoulos, C. (2021) The effect of sorptivity on cumulative infiltration. Water Science and Technology: Water Supply 21(2), pp. 606-614
  2. Sihag, P., Kumar, M., Sammen, S.S. (2021) Predicting the infiltration characteristics for semi-arid regions using regression trees. Water Science and Technology: Water Supply, 2021, 21(6), pp. 2583–2595
  1. Some words into the text should be corrected (e.g. “…the sand-covered slope exhibited a greater soil infiltration rate…”, where “exhibited” should be replaced with “showed”). So, an extensive review of the whole manuscript should be done, in order to correct English language expressions and spelling.

Specific comments

Tab. 2-1. You should add into the table the bulk density and the initial soil moisture. Did you measure the bulk density of the sand 1.3 g/cm3? Isn’t it too low for sand?

Fig 3-1, 3-2. According to theory, the stable infiltration rates should tend to hydraulic conductivity at saturation (Ks). Did you measure separately (maybe via the commonly used constant head method) the K­s?

Fig 3-5, 3-6 (why not named 3-3, 3-4?). Please define how you measured soil moisture. Are the y-axis values % w/w or %v/v? Please clarify. Same at Fig 3-7,…, 3-12.

Discussion section (1st paragraph): Do you mean porosity, or size of pores? These are inversely proportional quantities.

Conclusions: Conclusions are more like an abstract of the discussion, than conclusions. Basically, you found that infiltration rate is grater at sandy soil, which is well known. What are the new elements and the originality of this paper?  You must highlight the innovative points of your research, and maybe emphasize on some gaps that should be filled in the future.

Author Response

Dear editor and reviewers. Please see the attachment

Round 2

Reviewer 2 Report

I have read the revised manuscript and beleive that the corrections and revisions you made eleveted the quality of the paper. So, I think now it is ready for publication.